# Study for the Design of a Protocol to Assess the Impact of Stress in the Quality of Life of Workers

**DOI:** 10.3390/ijerph18041413

**Published:** 2021-02-03

**Authors:** Patricia Concheiro-Moscoso, Betania Groba, Francisco José Martínez-Martínez, María del Carmen Miranda-Duro, Laura Nieto-Riveiro, Thais Pousada, Cristina Queirós, Javier Pereira

**Affiliations:** 1CITIC, TALIONIS Group, Elviña Campus, University of A Coruña, 15071 A Coruña, Spain; patricia.concheiro@udc.es (P.C.-M.); f.martinezm@udc.es (F.J.M.-M.); carmen.miranda@udc.es (M.d.C.M.-D.); laura.nieto@udc.es (L.N.-R.); thais.pousada.garcia@udc.es (T.P.); javier.pereira@udc.es (J.P.); 2Faculty of Health Sciences, Oza Campus, University of A Coruña, 15071 A Coruña, Spain; 3Faculty of Psychology and Education Sciences, University of Porto, 4200-135 Porto, Portugal; cqueiros@fpce.up.pt

**Keywords:** work stress, occupational balance, occupational therapy, burnout, wearable technology, participatory health, Xiaomi Mi Smart Band 3

## Abstract

(1) Background: Work stress is one of the most relevant issues in public health. It has a significant impact on health, especially the development of mental disorders, causing occupational imbalance. There is a growing interest in the development of tools with a positive effect on workers. To this end, wearable technology is becoming increasingly popular, as it measures biometric variables like heartbeat, activity, and sleep. This information may be used to assess the stress a person is suffering, which could allow the development of stress coping strategies, both at a professional and personal level. (2) Methods: This paper describes an observational, analytical, and longitudinal study which will be set at a research center in A Coruña, Spain. Various scales and questionnaires will be filled in by the participants throughout the study. For the statistical analysis, specific methods will be used to evaluate the association between numerical and categorical variables. (3) Discussion: This study will lay the foundation for a bigger, more complete study to assess occupational stress in different work environments. This will allow us to begin to understand how occupational stress influences daily life activity and occupational balance, which could directly enhance the quality of life of workers if the necessary measures are taken.

## 1. Introduction

Work is one of the most significant areas of occupation in adulthood, and has great relevance at a social and personal level throughout life [1,2]. Work and working conditions have a significant impact on health [3]. Stressful situations occur in the worker when these conditions are altered by different factors (work overload, lack of support, work schedules, or lack of family reconciliation) [4,5,6], giving rise to a specific type of stress called work stress [7,8]. Work stress is defined as “the reaction that the individual may have to work demands and pressures that do not match his knowledge and skills, and that test his ability to cope with the situation” [9].

It is estimated that work stress affects three million workers worldwide [10], and that it represents between 50–60% of the cases of absenteeism and presenteeism [11,12]. In addition, several studies have reported that the health sector suffers the most from this type of problem [13,14,15]. In line with this data, occupational stress has become one of the most relevant problems for public and occupational health [16,17,18].

Work-related stress has important repercussions at the personal and health levels, but also at the social level [19]. Several studies have reported that work-related stress is a trigger for mental disorders (specifically depression [14,20,21], sleep disorders [22,23], heart and/or respiratory diseases, and physical or cognitive fatigue [24,25]. Moreover, if work-related stress is prolonged and worsened over time, it can lead to burnout in the worker [26,27]. Burnout was defined by Maslach, Schaufeli, and Leiter as “a prolonged response to chronic emotional and interpersonal stressors at work, defined by the dimensions of exhaustion and inefficiency” [28].

In addition to the physical or psychological effects that work stress can have, it is also related to alterations in the occupational balance [29], defined as “a balance of engagement in occupation that leads to well-being. The balance may be among physical, mental, and social occupations; between chosen and obligatory occupations; between strenuous and restful occupations; or between doing and being” [30,31]. Several studies refer to the fact that workers experiencing occupational stress are not able to relativize work and everyday occupations, prioritizing paid employment and leaving aside activities related to social interactions, household management, and rest or sleep [29,32,33].

Some studies have associated alterations in occupational balance with perceived stress [17,32]. Both concepts include the demands, needs, and emotions associated with daily life activities, in order to obtain satisfaction and well-being [32]. Studies such as that by Yu Yu et al. have emphasized the analysis of perceived stress [34,35], defined as “the individual’s ability to cope with stressful situations” [36]. Its analysis can help us to understand different predictors of stress, and prevent alterations that lead to an occupational imbalance [17].

The effects derived from work stress suppose important direct and indirect consequences for public health [16,37]. They generate an expense to the system due to high demand on health services, and the necessity of treatments that must be maintained in the long term like antidepressants or anxiolytics, which have a minimum duration of several months [38,39]. In addition, there is an economic cost since factors associated with occupational stress, such as low self-esteem, dissatisfaction, and low efficiency in the labor field, affect short-term productivity [40,41]. In the long term, the consequences can be aggravated by the occurrence of workplace accidents or temporary or permanent disabilities, with these representing one of the main causes of early retirement [42,43,44].

Due to the aforementioned, various organizations associated with health and work have the objective of promoting quality of life and well-being, as well as greater productivity, in the worker [11,45]. They consider of interest the creation of strategies and technological solutions for the early detection and control of factors and effects associated with work stress, as well as balance between personal and work life and the promotion of healthy habits [13,46].

Studies related to occupational stress have focused on the measurement of various physiological parameters for the detection of stress [47]. The most common biomarkers and indicators of stress are blood pressure and heart rate, which have been measured in these studies through the use of non-intrusive sensors to determine stress levels in workers [48,49]. However, these devices are generally uncomfortable for the user to carry for a long period of time, as is the case of belt-format pulsometers or portable electrocardiogram (EGC) devices [47,50]. Thus, it is necessary to find new sensors that are comfortable for the user, while providing information of equal quality [51].

In recent years, wearable devices for continuous, real-time monitoring of different parameters have been developed [52]. The measurement of parameters such as sleep and physical activity can be used to identify behavioral patterns of people when they are exposed to stressful situations [52]. Currently, the literature on the use of wearables for the measurement of occupational stress is focused on the development of automatic systems for the accurate assessment of stress, but less on the benefits of the study of stress at the social level [53,54,55]. However, studies such as that of Queirós et al. have already used Xiaomi wearable devices for public health purposes [56].

In this study, levels of work stress will be evaluated, along with how it influences the occupational balance and routine of workers. For this purpose, one of the most widely accepted wearables among the population will be used—the Xiaomi Mi Band 3—whose minute-by-minute measurements will be obtained through a data acquisition system developed by the research group. Likewise, specific scales and questionnaires will be used to measure the different aspects that influence quality of life.

### Objectives

The main objective of this study is to assess the work stress level and its influence on the quality of life of workers at a research center.

The specific objectives are:—To itemise the level of some parameters, such as activity, sleep, and heart rate.—To determine the level and impact of occupational stress in worker’s daily life through the information from wristbands and questionnaires.—To identify patterns of occupational functioning, physical activity, and sleep in these people.—To study the association between the identified patterns of occupational functioning, activity, and sleep, and the level of occupational stress and quality of life.

## 2. Materials and Methods

### 2.1. Study Design

This project will be a pilot study to determine the viability, sample size, cost, and duration of the study. Additionally, this project has been designed in order to “demonstrate that the planned measurements, the data collection instruments, and the data management system are feasible and effective” [57]. If this study proves to be successful, it will be performed with a larger sample from different work settings.

This study will be observational, analytical, and longitudinal. That is, in this study, different variables of the population under study will be observed and recorded without intervention, and with the aim of establishing causal associations between variables. It is considered longitudinal because the variables will be followed for six months, with continuous recording and monitoring of physical activity and the quality of sleep (wristbands). Variables related to work stress, quality of life, and perception of the quality of sleep and the level of physical activity (specific evaluation tools) will also be specifically measured [57]. This study protocol will follow the SPIRIT 2013 guidelines (See Appendix A) [58].

### 2.2. Participants and Settings

The study will be conducted in a research center in Galicia; a non-profit organization under private law whose aim is to contribute to the strengthening, empowerment, growth, and improvement of the competitiveness of the Galician information and communication technologies (ICT) sector. Sample selection will be performed through intentional sampling based on inclusion and exclusion criteria.

The participant inclusion criteria is: (a) performing management/administration/research tasks at the research center.

The participant exclusion criteria is: (a) close to retirement (5 years or less); (b) significant health condition complications that prevented active participation in the study; and (c) hypersensitivity in the skin or a recognized allergy to the material of the cases or straps of the wearable wristbands used as one of the measuring instruments of the study.

### 2.3. Recruitment Process

Entry into the study field will occur at the workplace where the project will be held. The participants will be recruited at the workplace, through an informative meeting that will be conducted at the facilities. In this meeting, the selection criteria for the study and the implications of participating will be presented, highlighting the total duration of participation and the mechanisms that will be followed at an ethical level to guarantee anonymity and the confidentiality of the data.

After the presentation of the main characteristics of the study to the workers attending the meeting, the information sheet will be given to any interested person to consider for as long as they wish, before making a decision regarding participation.

A week later, those workers willing to participate will be visited again to carry out the process of informed consent with the Principal Investigator (PI). In these individual meetings, possible doubts or queries will be resolved, the established selection criteria will be checked for each person, and the informed consent document will be signed if applicable.

### 2.4. Justification of Sample Size

This pilot study is being suggested in order to determine the feasibility, time, and cost required for a subsequent study, as well as to demonstrate that the selected measurements and instruments are feasible and effective, and to determine, based on the results, the optimal frequency, intensity, and duration of the study.

Likewise, by carrying out this pilot study, we will gain information about the standard deviation and the proportion of participants with the specific characteristics needed for this study, which will allow us to carry out the calculation of the sample size.

### 2.5. Outcomes

The primary outcome will be to determine the influence of stress and anxiety on the worker. The secondary measures will be: (a) sleep recording and physical activity tracking with the Xiaomi Mi Band 3; (b) quality of life self-perception with the EuroQol 5D-5L (EQ 5D-5L) scale; (c) determination of sleep habits by using the Pittsburgh Sleep Quality Index (PSQI); (d) anxiety self-perception by using the State-Trait Anxiety Inventory (STAI); (e) stress self-perception by using the Perceived Stress Scale-10 (PSS-10); and (f) influence of stress on daily functioning, by using a questionnaire designed by a work stress psychologist professional in consensus with the socio-sanitary professionals of the group.

### 2.6. Data Collection and Management

Prior to any other data, socio-demographic features of the participants considered as relevant will be collected using a record sheet of our own elaboration. This questionnaire includes questions regarding age, gender, marital status, residential environment, cohabitation unit, educational level, profession, socio-economic level, working hours, overtime working hours, level of perceived stress, other non-work-related stress factors, actions oriented to stress reduction, and medication intake.

Data will be collected from different sources (Figure 1). For biometrical data, a Xiaomi Mi Band 3 will be worn by all the participants for six months to measure their physical activity (steps), sleep (hours of sleep and sleep stages), and heart rate (bpm). A self-developed data collection system will be installed in one of the computers of the research center, capturing the data from the wristbands via Bluetooth every time they walk nearby and storing it in an Structured Query Language (SQL) database [59]. When the data is analyzed, it will be retrieved by the SQL database in which it was stored. The data will be exported through CSV in a detailed or accumulated way for analysis in any statistical program.

The Research Electronic Data Capture Consortium (REDCap), a safe web application to create and manage online surveys and databases [60], will be used to digitally create the questionnaires and tests explained below, which will be sent to the participants to fill out.

In this study, participants will complete a set of assessment tools and a self-made questionnaire associated with work overload, sleep, and physical activity. These assessment tools will be filled in at the beginning and end of the study, while the self-made questionnaire will be answered throughout the study. This information is shown in Table 1.

The scales to be completed are: (1) the EQ 5D-5L, which evaluates mobility, personal care, daily activities, pain/discomfort, anxiety/depression, and global status at the moment of assessment [61], (2) the PSQI, which analyzes the quantity, quality, duration, latency, and efficiency of sleep [62], (3) the STAI, which evaluates the presence of anxiety [63], and (4) the PSS-10, which measures the degree to which, during the last month, people have felt annoyed or worried or, on the contrary, have felt confident in their ability to control their personal problems [64].

Finally, a self-made questionnaire was developed considering the results of a prior poll that assessed different factors potentially related to stress. Those items that were directly related to stress were chosen for the questionnaire. This final questionnaire was agreed on by a stress psychologist professional. It is composed of three daily and four weekly items to answer (Appendix B), related to stress, work commitment and frustration, sleep, physical activity, and occupational balance. All items in the weekly and daily questionnaire are scored from 0 to 5 (0 = Nothing; 1 = Very little; 2 = Little; 3 = Some; 4 = Quite a lot; 5 = A lot).

### 2.7. Data Analysis

The statistical treatment of the data will be performed with the statistical package IBM SPSS Statistic version 22 (IBM, Chicago, IL, USA). The significance level for hypothesis testing is set at 5%. The numerical variables will be expressed as the mean (M) and standard deviation (SD), including range, minimums, and maximums, while the categorical variables will be shown with their absolute frequency and valid percentage. In addition to the simple description of the data and variables studied, inferential analyses will be conducted in order to determine possible significant relationships between the variables of the study, or to allow the contrast of hypotheses. The Kolmogorov–Smirnov test will be applied to check that the variables behave as a normal distribution [65]. If not, non-parametric tests will be used in the subsequent analysis. The association of numerical variables will be analyzed with the Pearson or Spearman’s Rho correlation test, depending on the distribution of the sample. To assess the association of categorical variables, the Chi-square test will be applied, or the likelihood ratio in the case that the frequencies observed are less than 5%. In the case of the association between quantitative and qualitative variables, the comparison of means will be made with Student’s *t*-test and analysis of variance (ANOVA) or Mann–Whitney and Kruskal–Wallis’ U test, as appropriate [66]. Finally, and in order to determine whether there are significant and relevant differences between the results of the evaluations performed (initial, intermediate, and final), the Wilcoxon test will be applied [66]. Data analysis includes the cleaning or pre-processing, description, and processing of the stored data. The final objective of these steps is to obtain information for decision making. During pre-processing, erroneous data are removed or corrected to avoid bias in the results. Subsequently, a descriptive statistical study which summarizes certain relationships and distributions in a simple way will allow us to choose which processing strategy to adopt.

### 2.8. Ethics and Dissemination

This study protocol was approved by the A Coruña-Ferrol Research Ethics Committee under the number 2019/249, with the date of 24 April 2019. In addition, this protocol was registered in ClinicalTrials under the identifier NCT04584021, on 12 October 2020. It is available at https://clinicaltrials.gov/ct2/show/NCT04584021.

In case any change in the protocol is needed, this will be communicated to the Research Ethics Committee with the assigned reference number. These modifications will be updated in the Clinical Trials register.

The impact that the use of the procedures defined in the project may have on the protection of the personal data of the participants will be evaluated, emphasizing that such data will never leave the information system of the “Tecnología Aplicada a La Investigación en OcupacióN, Igualdad y Salud” (TALIONIS) Group, following the regulations of the Spanish and European Organic Law on the protection of personal data [67,68]. Data confidentiality and anonymity will be maintained through pseudonymization techniques. Once the study is completed, the data will be stored anonymously with the previous authorization of the participants.

The results of the questionnaires and assessment tools will be administered, coded, and managed through the software of REDCap. This platform complies with data protection regulations and stands out for being a widely used tool in clinical trials at an international level. The PI will have access to all the identification data of the participants and all the data collected during the study; the rest of the research team will have access to the data of the participants without identification through the REDCap platform, which allows the capture and consultation of the data anonymously.

The results related to the data of the wearable wristbands will be automatically captured through the wearable wristband data acquisition module. This platform was made following software design standards to guarantee the security and anonymization of the data during capture. It was developed in the Python programming language, along with the necessary libraries for the capture and storage of data in an anonymous way. The data is stored in an SQL server database that, together with the anonymization mechanisms and strategies created by the research team, has data masking mechanisms. In addition, it has limited access to data and security and access control mechanisms. Regarding Bluetooth technology, each bracelet has a unique internal key, generated at random, which is automatically changed. All these software resources are located in a small computer that will be placed in the environment of the center. This computer does not have input and output mechanisms to avoid any person outside the computer trying to access it. The PI will have access to the data related to the identification of the participants, and to the relationship of each participant with the internal number of each wristband. Once the study is finished, this data relationship will be eliminated.

## 3. Discussion

This pilot study may allow us to design a broader study with more participants in future. Stress is increasingly present in our society, and despite being historically understood as an expected effect derived from the mere activity of working, in recent years it has started to be considered as detrimental, but not unavoidable [69,70]. This kind of study could help with the control and detection of stress, influencing positively its reduction and improving the quality of life of workers [37]. In particular, if this study is successful, it would allow the evaluation of occupational stress by using cheap devices and easy-to-understand questionnaires that can be used by any person without any specific training. It would be possible to establish the influence that working stress has on the daily routines and occupational balance of workers.

In addition to the worldwide increase in stress, the emergence of coronavirus disease 2019 (COVID-19) has meant a significant worsening of the mental health of a large part of the population. Stress, along with anxiety and depression, have spread or become more severe during the pandemic in the general population [71]. Job instability due to unforeseeable lockdowns and restrictive measures leads to increased stress in the population, which along with precarious jobs, generates long-term uncertainty that can further increase general stress levels [72,73].

## 4. Conclusions

This project contributes to know the influence of occupational stress on the quality of life in a work environment. Therefore, it is intended to observe some parameters (activity, sleep, and heart rate) recorded through Xiaomi Mi Band 3. In addition, it aims to know about the level of work stress in diverse aspects of daily living, using various online self-report questionnaires in different periods of time. This data obtained from the questionnaires and wristbands will allow to determine the level of stress and its influences on quality of life during and after work.

The increasing use of portable devices such as smartphones or wearable devices encourages to obtain real-time biomedical data available for people, without going to health resources. Thus, this project considers as innovative that all measurements are registered online so that the workers can have an insight into their level of stress, the different stressful factors, and other parameters related to their activity, sleep, and heart rate, which influences their quality of life and work activity. It may help them to be more conscientious about their situation, so they can decide to make changes in their daily routines and habits.

Finally, considering the current pandemic situation, the online registration of all measurements will provide a follow-up of the participants without the need for face-to-face meetings.

## Figures and Tables

**Figure 1 ijerph-18-01413-f001:**
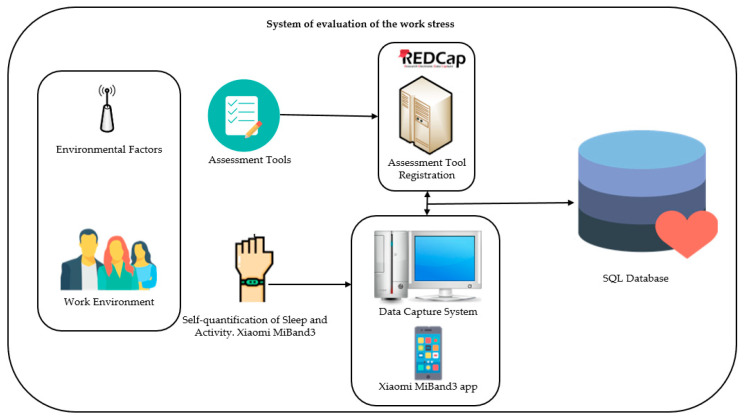
Registration and capture of biometric data and the assessment process.

**Table 1 ijerph-18-01413-t001:** Measuring assessment tools.

Instrument	Data	Delivery Phase
EuroQol-5D-5L (EQ 5D-5L)	Quality of life	Beginning, follow-up, and completion
Pittsburgh Sleep Quality Index (PSQI)	Sleep habits	Beginning, follow-up, and completion
State-Trait Anxiety Inventory (STAI)	Anxiety	Beginning, follow-up, and completion
Perceived Stress Scale-10 (PSS-10)	Stress	Beginning, follow-up, and completion
Stress Questionnaire	Work stress factors and daily functioning	Daily (3 items) and weekly (4 items)

## Data Availability

Once the data collection process is finished and these have been coded, structured, and analyzed, these data will be provided, after the Spanish Data Protection Agency’s consent, to any researcher that contacts TALIONIS group Principal Investigator, Javier Pereira (Javier.pereira@udc.es).

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
