# Peer review of "Study for the Design of a Protocol to Assess the Impact of Stress in the Quality of Life of Workers"

_ijerph, 2021, doi:10.3390/ijerph18041413_

Round 1
Reviewer 1 Report
Dear author, you performed a pilot study about the feasibility of the assessment of stress level and quality of life in a worker population.
I appreciate that you follow the SPIRIT guidelines and the methods and search strategy is well defined.
However, in my opinion, the introduction should be more succinct highlight less information important for the paper.
Moreover, inclusion criteria could be better explained. English form should be improved
Author Response
Dear reviewer,
We attach the responses to your comments.
Point 1: The introduction should be more succinct highlight less information important for the paper.
Response 1: Thank you for reviewing the manuscript. According to your comment and the other reviewers' comments, the authors have tried to specify the information to make the introduction more concise.
Point 2: Inclusion criteria could be better explained.
Response 2: Thank you for your suggestion. We have improved the explanation of the criteria for inclusion of the manuscript: "a) performing management/administration/research tasks at the research center.". You can see it marked in yellow on page 3, lines 133-134.
Point 3: English form should be improved
Response 3: Thank you for your comment. The authors sent the manuscript to the English editor of IJERPH. Please see the attachment.

Reviewer 2 Report
Dear authors, thank you for possibility of reading your interesting study protocol. The issue of work stress is highly important and fits into the IJERP’s scope and aims. The paper describes a longitudinal pilot study protocol (6 month period) with the use of observational data (biometrical data measured via wristbands) and questionnaires (mostly standardized).
The strength of this project is a mixed method of observational data and self-report questionnaires. This is of high importance for accuracy of evaluation of stress levels related to work environment. The measurement of the heart rate during work and comparison to heart rate after work, gives an opportunity for in-depth analyses to the longitudinal effects of work stressors on quality of life.
Considering the ongoing pandemic, it should be highlighted, that all measurements will be undertaken online, without the need of in person meetings in the future study. The paper is well structured.
I consider this project as relevant and sufficient.
Nevertheless, there are a few points that should be adjusted or elaborated.
- Title, line.3: Quality of Life is a set phrase. Please, consider this change in your title
- Introduction:
- the paper deals with the issue of stress related to work. It will be measured by both, heart rate (bmp) as observational data and perceived stress questionnaire. At least one paragraph should be dedicated to the theoretical basis of perceived stress.
- also, the definition of work stress is derived from a conference presentation [9] and not the scientific paper. It should be change.
- there is a strong emphasis on the occupational balance in the introduction (lines 56-63), and it is listed as a crucial result of the study (line:94), whereas it is measured only by one item. The introduction shows how crucial it is for the study, the worry is whether you will deliver based on this simplistic measurement. Could you please elaborate this issue.
- Objectives: The objectives are well emphasized, in the correct part of the paper. The main aim is clear whereas specific objectives need further elaboration. Specific objective number 1.: How are you going to verify the accuracy of wristbands accuracy? Is it the goal of your study? How are you going to measure it?
- The main issue in the proposed study is occupational stress. It is mentioned in the introduction, that the main predictors are related to conditions of work, yet, the working conditions are roughly mentioned in the measurements description (working hours and overtime). There are many other aspects of work environment, like physical factors (e.g., confined spaces, temperature, lighting, noise, vibration), not to mention technostress, that is the main concern in the ICT sector (Pflünger et al., 2020). Are those variables going to be addressed? Could you include at least some of them in the first measurement while gathering sociodemographic data? This is particularly important, if you willing to use it within a larger sample of diverse employee groups that can experience different work environment. I would also encourage you to ask about possibility of coaching and mentoring in the company, as those are relevant variables related to stress at work. Otherwise, many important variables (stress reduction technical, medication intake) have already been included into the study design.
- Most of the measurements used in this study are standardized except for Stress questionnaire (work stress factors and daily functioning) “composed of 3 daily and 4 weekly items” (line 200). It is vaguely presented in lines 166-167. Nevertheless, since the new measurement has been created, it should be elaborated on the questionnaire premises and the description of proposed items would be expected.
- line 170: what do you mean by “a questionnaire made by the group”? Do you mean that the research group will participate in creating the specific measurement?
- It is assumed in the pilot study that the observational data will be gathered via Bluetooth directly to the SQL base. Nevertheless, it is assumed, the further study is planned (based on the results from a pilot study), that will address the larger number of participants. Could you elaborate how the data will be retrieved from participants in the further study (probably within the employees from outside the Galician research centre)?
Author Response
Dear reviewer,
We attach the responses to your comments.
Point 1: Title, line 3: Quality of life is a set phrase. Please, consider this change in your title.
Response 1: Thank you for your comment. We agree with you, so we changed the title to: “Study for the design of a protocol to assess the impact of Stress in the Quality of Life of Workers” You can see the change, marked in yellow, on page 1.
Point 2: The paper deals with the issue of stress related to work. It will be measured by both, heart rate (bmp) as observational data and perceived stress questionnaire. At least one paragraph should be dedicated to the theoretical basis of perceived stress.
Response 2: Thank you for your appreciation. According to you, the authors consider that this concept is important. So, the authors have added a paragraph on perceived stress in the introduction. You can see it, highlighted in yellow (page 2, lines 64-69).
Point 3: also, the definition of work stress is derived from a conference presentation [9] and not the scientific paper. It should be change.
Response 3: Thank you for your comment. The authors changed and marked in yellow reference 9 (page 1 and 9, lines 44 and 332).
Point 4: there is a strong emphasis on the occupational balance in the introduction (lines 56-63), and it is listed as a crucial result of the study (line:94), whereas it is measured only by one item. The introduction shows how crucial it is for the study, the worry is whether you will deliver based on this simplistic measurement. Could you please elaborate this issue.
Response 4: Thank you very much for your comments and suggestions. The authors understand your concern about this issue. According to scientific evidence, occupational stress mainly influences occupations such as sleep, social participation, and the management of different activities. For this reason, the authors consider that we can know the preliminary situation of the participants' occupational balance through the information obtained from the wristbands and some items (sleep, activity, task overload, and occupational balance) of the weekly questionnaire.
Point 5: Objectives: The objectives are well emphasized, in the correct part of the paper. The main aim is clear whereas specific objectives need further elaboration. Specific objective number 1.: How are you going to verify the accuracy of wristbands accuracy? Is it the goal of your study? How are you going to measure it?
Response 5: Thank you for your comments. The authors agree that specific objectives can be clearer. Therefore, specific objectives have been changed. Specifically, the authors have changed specific objective 1, which the reviewer has highlighted. You can see them, marked in yellow, on page 3, lines 107-112.
According to the reviewer, the authors can't verify the accuracy of wristbands, because it is necessary to compare this device with a gold standard dispositive. This is an aspect that is being worked on in another study. Thus, the specific objective will be to know the diary level of some parameters (activity, sleep, and heart rate) through the Xiaomi Mi Band 3's information.
Point 6: There are many other aspects of work environment, like physical factors (e.g., confined spaces, temperature, lighting, noise, vibration), not to mention technostress, that is the main concern in the ICT sector (Pflünger et al., 2020). Are those variables going to be addressed? Could you include at least some of them in the first measurement while gathering sociodemographic data? This is particularly important, if you willing to use it within a larger sample of diverse employee groups that can experience different work environment. I would also encourage you to ask about possibility of coaching and mentoring in the company, as those are relevant variables related to stress at work. Otherwise, many important variables (stress reduction technical, medication intake) have already been included into the study design.
Response 6: The authors appreciate very much for these suggestions. At first, the variables such as temperature, spaces, light, or noise were going to be measured. But, as they were not seen as achievable by the center and the research group. The decision was made not to take them into account.
On the other hand, although the study will be carried out in an ICT research center, the participants will be specific people associated with administration, management, or research issues. Therefore, the technostress has not been contemplated for this research. But the authors consider that these variables are important, and they will take them into account in the future.
Point 7: Most of the measurements used in this study are standardized except for Stress questionnaire (work stress factors and daily functioning) “composed of 3 daily and 4 weekly items” (line 200). It is vaguely presented in lines 166-167. Nevertheless, since the new measurement has been created, it should be elaborated on the questionnaire premises and the description of proposed items would be expected.
Response 7: Thank you for your appreciation. The authors agree with your suggestion. An appendix has been added with the description of the proposed items (page 8). As well as, the authors have explained the punctuation of these items (See on page 5, lines 199-203).
Point 8: line 170: what do you mean by “a questionnaire made by the group”? Do you mean that the research group will participate in creating the specific measurement?
Response 8: Thank you for your comment. The sociodemographic questionnaire is not a questionnaire created by the group, but a record sheet with different variables that the authors have elaborated. Therefore, we have modified this paragraph and highlighted in yellow this modification (See on page 4, line 170).
Point 9: It is assumed in the pilot study that the observational data will be gathered via Bluetooth directly to the SQL base. Nevertheless, it is assumed, the further study is planned (based on the results from a pilot study), that will address the larger number of participants. Could you elaborate how the data will be retrieved from participants in the further study (probably within the employees from outside the Galician research centre)?.
Response 9: Thank you for your appreciation. There will be a Bluetooth device in all workplaces. This device collects the data stored in the device and dumps it into a SQL database. To recover this data, the authors will access the SQL database and obtain the detailed or accumulated data in a CSV file, which can be used in any statistical program. This information is highlighted in yellow on page 4, lines 178-180.
Finally, the authors want to comment you that the document was sent to the English editing service of the IJERPH. Please, see the attachment.

Reviewer 3 Report
This study is a study that proposed a research plan. A protocol that measures the life quality of a worker in real-time through a wearable device is proposed.
I feel that this study is not yet worth being published in its current state. The value of this study can be judged only when the results of the pilot project are provided together with the research proposal.
In particular, although this study is a study protocol, it seems necessary to present the purpose of the study in it, identify problems in the past, and compare various alternatives. In the absence of these points, the value of this study is not great, and publication cannot be recommended unless the pilot study is completed and any analysis process is performed.
Author Response
Dear reviewer,
We attach the responses to your comments.
Point 1 : I feel that this study is not yet worth being published in its current state. The value of this study can be judged only when the results of the pilot project are provided together with the research proposal.
In particular, although this study is a study protocol, it seems necessary to present the purpose of the study in it, identify problems in the past, and compare various alternatives. In the absence of these points, the value of this study is not great, and publication cannot be recommended unless the pilot study is completed and any analysis process is performed.
Response 1: Thank you very much for your suggestions and comments. According to your comments, the authors have tried to show these points in the manuscript. Thus, the purpose of the study can see on page 3, lines 98-102.
Your comment about the inclusion of the analysis process in the manuscript is interesting. However, given the nature of the manuscript and following the Spirit guidelines, the authors can only present the design of a pilot study. This pilot study aims to know the viability of the study and improve it if it' would be necessary. In addition, we attached the certificate of the manuscript's English editing.

Round 2
Reviewer 3 Report
This study cannot be recommended for publication in its current state.Although this study is in the form of a study protocol, the background and the rationale for the creation of the protocol should be provided, but they are missing.
Surveys on past literatures are also very limited.
Even in the Revised version, the points pointed out were not modified at all.
Author Response
This study cannot be recommended for publication in its current state. Although this study is in the form of a study protocol, the background and the rationale for the creation of the protocol should be provided, but they are missing. Surveys on past literatures are also very limited. Even in the Revised version, the points pointed out were not modified at all.
Thank you for this suggestion. If the editor considers, the authors may conduct a new, exhaustive literature about work stress and wearables devices. However, as we stated in the answer to the reviewer 3, there is short literature on this topic, so it would be helpful if the reviewer specifies of which information we are lacking. The authors include the following references 48-57 of the revised version of the manuscript about the past literature. See on page 2, lines 83-96.